# Local Landscapes, Evolving Minds: Mechanisms of Neighbourhood Influence on Dual-State Mental Health Trajectories in Adolescence

**DOI:** 10.3390/ijerph22060951

**Published:** 2025-06-17

**Authors:** Christopher Knowles, Emma Thornton, Kathryn Mills-Webb, Kimberly Petersen, Jose Marquez, Sanja Stojiljković, Neil Humphrey

**Affiliations:** 1Manchester Institute of Education, The University of Manchester, Manchester M15 6JA, UK; emma.thornton@manchester.ac.uk (E.T.); kathryn.mills-webb@manchester.ac.uk (K.M.-W.); jose.marquez@manchester.ac.uk (J.M.); neil.humphrey@manchester.ac.uk (N.H.); 2School of Education, The University of Leeds, Leeds LS2 9JT, UK; k.petersen@leeds.ac.uk; 3Faculty of Philosophy, The University of Belgrade, 11000 Belgrade, Serbia; sanja.stojiljkovic@f.bg.ac.rs

**Keywords:** adolescents, environmental psychology, internalising symptoms, latent variable modelling, life satisfaction, neighbourhood effects, wellbeing

## Abstract

Neighbourhood variation in socioeconomic deprivation is recognised as a small but meaningful determinant of adolescent mental health, yet the mechanisms through which the effects operate remain poorly understood. This study used #BeeWell survey data collected from adolescents in Greater Manchester (England) in 2021–2023 (life satisfaction: N = 27,009; emotional difficulties: N = 26,461). Through Latent Growth Mixture Modelling, we identified four non-linear trajectories of life satisfaction (Consistently High (71.0%), Improving (8.7%), Deteriorating (6.3%), and Consistently Low (13.9%); entropy = 0.66) and three non-linear trajectories of emotional difficulties (Low/Lessening (53.7%), Sub-Clinical (38.3%), and Elevated/Worsening (8.0%); entropy = 0.61). Using a multi-level mediation framework we assessed (1) whether neighbourhood deprivation predicted trajectory class membership and (2) the extent to which effects of deprivation operate through aspects of Community Wellbeing, as measured by the Co-op Community Wellbeing Index (CWI). Greater deprivation increased the odds of following Deteriorating (OR = 1.081, [1.023, 1.12]) and Consistently Low (OR = 1.084, [1.051, 1.119]) life satisfaction trajectories and reduced the odds of following a Sub-Clinical emotional difficulties trajectory (OR = 0.975, [0.954, 0.996]). Mediation analyses revealed that the effects of deprivation on Consistently Low life satisfaction partially operate through Equality (*ab* = 0.016, [0.002, 0.029]) and Housing, Space, and Environment (*ab* = −0.026, [−0.046, −0.006]). Further indirect effects were observed for Housing, Space, and Environment, which reduced likelihood of Sub-Clinical emotional difficulties for those living in deprived neighbourhoods (*ab* = −0.026, [−0.045, −0.008]). The findings highlight the distinct effects of neighbourhood deprivation on affective and evaluative domains of adolescent mental health and the protective effect of housing and related environmental factors in disadvantaged contexts, advancing our understanding of the mechanisms underpinning neighbourhood effects on dual-state adolescent mental health.

## 1. Introduction

Ecological system theories recognise the role of environmental context in adolescent wellbeing [1,2]. The research has consistently highlighted the interconnectedness of constructs such as peer relationships, school environments, and family dynamics, and their collective bidirectional relationship with adolescent mental health [3,4,5]. In particular, studies focused on the influence of a young person’s surrounding neighbourhood have revealed that environmental factors and perceptions of their local community can play a crucial role in shaping mental health outcomes [6,7]. There is evidence that neighbourhoods influence key aspects of developmental progression, including identity formation, autonomy-seeking, and the establishment of strong social connections [6,8]. Due to the universal nature of neighbourhood exposure, even small effects explaining just 2–3% of the variation in outcomes can have substantial population-level consequences for educational attainment, career prospects, and health behaviours [6,7]. Given the cumulative effects of environmental influences, neighbourhood characteristics—ranging from infrastructure and proximity to green space, to social cohesion, connectivity, and safety—should be considered when developing strategies to improve adolescent mental health [3,4,9].

### 1.1. Mechanisms of Neighbourhood Influence

Neighbourhood differences are commonly explained in terms of area-level socio-economic deprivation. In the UK, this is typically measured via the Index of Multiple Deprivation (IMD), a centrally developed indicator of social disparity. Associations with IMD are frequently attributed to the cumulative impact of material hardship, limited access to essential services, housing instability, and exposure to chronic stressors such as violence, inadequate infrastructure, and financial insecurity. That said, neighbourhood effects extend beyond these factors to the broader ecological context in which adolescents grow up. Social connectedness within a neighbourhood—often discussed under the umbrella of social capital—plays a critical role in supporting adolescent mental health. Adolescents living in neighbourhoods characterised by high levels of trust, mutual aid, and social connection among neighbours tend to report better mental health outcomes, even after adjusting for deprivation [10]. The mechanisms include increased social support, reduced exposure to violence, and the availability of role models and prosocial norms. Conversely, social fragmentation, low collective efficacy, and a perceived lack of belonging can increase the risk of loneliness, marginalisation, and ultimately problematic trends in population mental health [10,11]. Perceptions of safety and exposure to crime have also been independently associated with adolescent mental health outcomes. Those who feel unsafe in their local neighbourhood may be less likely to engage in outdoor physical activity, socialise with peers, or access local amenities [11]. Evaluating how factors such as equality, social capital, and perceptions of safety and community support (henceforth termed aspects of community wellbeing) function as mediating pathways through which the effects of neighbourhood deprivation operate would provide valuable insight to public health agencies in England and aid the development of policies aimed at mitigating the adverse effects of deprivation on adolescent mental health outcomes.

### 1.2. Dual-State Approaches to Mental Health

A dual-state approach to mental health shows equal appreciation for both positively and negatively orientated components. In particular, Keyes’ two-continua model defines mental health as ‘more than the absence of a disorder’ and posits that in order to flourish, one must not only experience minimal symptoms of ill health, but actively demonstrate positive signs of wellbeing (e.g., positive emotions, self-esteem, and life satisfaction) [12]. Dual-state approaches offer holistic perspectives, incorporating social, psychological, and emotional domains of wellbeing to provide a more comprehensive assessment of overall wellbeing. Moreover, mental health and ill health are not always subject to the influence of a mutual set of predictors (i.e., factors that affect mental health do not necessarily affect ill health) further emphasising the need to assess these factors in tandem, but as independent and unique constructs [13,14]. Such a viewpoint expands our understanding of mental health beyond the traditional dichotomy of pathology and health.

### 1.3. Person-Centred Analysis

Social scientists are increasingly turning to person-centred analyses to better understand the unique characteristics of their target population. Unlike variable-centred approaches, which assume a homogeneous population (with deviations from the sample mean attributed to a chosen variable of interest), person-centred approaches leverage the natural heterogeneity that exists within populations, using predictor variables to estimate the likelihood of exhibiting unobserved (i.e., latent) clusters of characteristics or traits. Latent Growth Mixture Modelling (LGMM) examines clustering longitudinally by identifying individuals following similar trajectories over time who are quantitatively and qualitatively distinct from those in other classes (e.g., an improving class versus a deteriorating class), facilitating assessments of both within-group homogeneity and between-group heterogeneity [15,16]. Given that adolescence is a particularly transformative phase of development, changes (or lack thereof) in self-reported life satisfaction and emotional difficulties naturally occur at different rates, making person-centred approaches particularly illuminative and essential for understanding how the social and demographic features of the neighbourhood context impact mental health over time [17].

### 1.4. The Current Study

This study advances the evidence base by examining the relationship between neighbourhood deprivation and adolescent mental health and wellbeing. First, unlike most previous research, which has generally taken a variable-centred approach that overlooks population heterogeneity, we used a mixture modelling framework to identify unobserved subgroups of adolescents following latent longitudinal mental health trajectories in an area of research dominated by cross-sectional studies. Second, we employed a multi-level mediation framework, investigating the mechanisms through which neighbourhood deprivation operates where *x* = neighbourhood deprivation, *m* = aspects of community wellbeing, and *y* = latent trajectory class membership. Few have explored the mechanisms through which neighbourhoods influence adolescent mental health [6], with most focusing on individual or family-level factors such as family functioning and peer relationships. Fewer still have assessed neighbourhood effects using a multi-level approach whereby individual-level factors are modelled on the within level, and neighbourhood factors modelled at the between level. An exception is Putra et al. (2024) [18], which examined how neighbourhood characteristics mediated the association between socioeconomic status and mental health in children aged 5 to 9. We extend this work by investigating how community-level factors mediate these relationships in adolescents aged 12 to 15, a critical developmental period when young people begin to spend more independent time in their local neighbourhood. Furthermore, while Putra et al. (2024) [18] assessed overall mental health using total Strengths and Difficulties Questionnaire scores (Goodman, 1997) [19], we employed a dual-factor approach to distinguish how community wellbeing indicators function as either promotive or risk factors for positively and negatively orientated domains of mental health.

### 1.5. Aims and Hypotheses

The aims of the present study were three-fold: (1) determine the latent trajectories of life satisfaction and emotional difficulties in a large sample of adolescents from Greater Manchester, England; (2) compare the probability of latent trajectory class membership across Greater Manchester (GM) neighbourhoods according to neighbourhood-level deprivation; and (3) determine the extent to which the effects of neighbourhood-level deprivation on adolescent life satisfaction and emotional difficulties trajectories operate through different aspects of community wellbeing.

RQ1: Are Quantitatively and Qualitatively Distinct Mental Health Trajectories Observable in Adolescence?

As the enumeration phase is exploratory by nature, we did not presume to find a specific number of latent classes. Likewise, due to the heterogeneity that exists within adolescent self-reported mental health, it was plausible we would identify a range of trajectories including high, low, improving, deteriorating, fluctuating, and non-linear growth. However, we predicted that the proportional distribution of the sample would be most heavily weighted toward a class characterised by fairly stable, moderate levels of life satisfaction (H^1a^) and fairly stable, low levels of emotional difficulties (H^1b^).

RQ2. Are There Neighbourhood Differences in the Likelihood of Exhibiting Different Trajectories of Life Satisfaction and Emotional Difficulties?

We predicted that trajectory class membership would vary across neighbourhoods (H^2a^), with greater neighbourhood deprivation increasing the probability that adolescents followed less favourable life satisfaction (H^2b^) and emotional difficulties (H^2c^) trajectories.

RQ3. Do Associations Between Neighbourhood Deprivation and Trajectories of Life Satisfaction and Emotional Difficulties Operate through Aspects of Community Wellbeing?

We predicted that the associations between neighbourhood deprivation and adolescent dual-state mental health trajectories (life satisfaction and emotional difficulties) operate through several aspects of community wellbeing (i.e., neighbourhood-level indicators of social disparity and social drivers of wellbeing, such as housing, equality, and education) with more favourable scores for community wellbeing linked to more favourable mental health trajectories. The hypotheses for each aspect of community wellbeing are presented in Table 1.

## 2. Materials and Methods

### 2.1. Data

#Beewell is a hybrid population cohort study comprising a truncated longitudinal study in which participants are tracked, with annual data points from the age of 12 to 15 (i.e., from Year 8 to 9 to 10 of secondary school; Sample 1), and a serial cross-sectional study comprising annual data points for participants aged 14–15 (i.e., those in Year 10 of secondary school at a given timepoint; Sample 2) [20]. Together, the #BeeWell dataset and linked administrative data provided by GM local authorities and partner organisations comprise a rich source of individual-level (e.g., life satisfaction, hope and optimism, and emotional difficulties) and neighbourhood-level (e.g., socio-economic deprivation and housing) data on adolescent wellbeing. The present study used the first three-years of data (2021 to 2023) from Sample 1 in GM, England, beginning when the participants were in Year 8 of secondary school. For the current study, aspects of community wellbeing were quantified using the Co-op Community Wellbeing Index (CWI), which is described in detail below.

### 2.2. Participants

All adolescents from Sample 1 providing either life satisfaction or emotional difficulties data at one or more timepoints were eligible for inclusion on a per model basis (N = 27,009 for life satisfaction; N = 26,461 for emotional difficulties). Descriptive statistics for the underlying samples used to establish the analytical models are provided in Table 2. Appendix A comparing the sample’s demographic information against that of GM and England is also provided (Appendix A. In short, the analytical sample closely reflects the GM population in sex, ethnicity, the proportion speaking English as an additional language, and special educational needs, but includes somewhat fewer adolescents eligible for free school meals. Compared to the national population, it aligns in most respects but has a slightly lower proportion of White British adolescents and a higher proportion of young people eligible for free school meals (reflecting differences between the composition of the populations of GM and England). 

### 2.3. Measures

#### 2.3.1. Life Satisfaction

The Office for National Statistics single-item measure of life satisfaction was presented at each timepoint [21]. The item asks, “Overall, how satisfied are you with your life nowadays?” and is rated on a Likert scale ranging from 0 (not at all) to 10 (completely) where scores ≥ 7 are considered high. The item is currently used by the UK Office for National Statistics to quantify life satisfaction and can be operationalised to offer an alternative measure of societal progress beyond simply GDP, and the economic impact of disparity/change in life satisfaction over time through the application of WELLBYs [22]. A WELLBY (or Wellbeing-Year) is defined as one point on our life satisfaction item for one individual for one year. In the UK, as of 2021, the social production cost of one WELLBY has a proposed central value of GBP 13,000 per person. As an illustrative example, assuming linear change, a rise in life satisfaction from 6.5 to 7.0 from 2021 to 2022 (i.e., a 0.25-point rise in average life satisfaction relative to the starting value) equates to a roughly GBP 3250 increase in GDP per capita (0.25 × GBP 13,000). The same rise across two years equates to an annual increase of GBP 1625 in GDP per capita.

#### 2.3.2. Emotional Difficulties

The 10-item emotional difficulties subscale of the Me and My Feelings Questionnaire (M&MF) was presented at each timepoint [23]. The items probe emotional difficulties commonly reported by young people (e.g., *I feel lonely*, *I am unhappy*, and *I worry a lot*) and are scored on a three-point scale where 0 = *never*, 1 = *sometimes*, and 2 = *always*. To aid model convergence, emotional difficulties were quantified as the total score across all 10-items with a range of 0 to 20 with scores ≥ 12 representing elevated symptoms. The M&MF has been validated for use in children aged 8+ and had a high level of convergent and discriminant validity [24]. In the current sample, Cronbach’s α ranged from 0.87 to 0.90, indicating a high level of internal consistency.

#### 2.3.3. The Co-Op Community Wellbeing Index

The Co-op Community Wellbeing Index (CWI) captures several aspects of community wellbeing, encompassing nine domains that sit within the three core pillars of relationships (1. *Relationships and Trust*; 2. *Equality*; 3. *Voice and Participation*), place (4. *Health*; 5. *Education and Learning*; 6. *Economy, Work, and Employment*), and people (7. *Culture, Heritage, and Leisure*; 8. *Space, Housing, and Environment*; 9. *Transport, Mobility, and Connectivity*) [25]. Each domain is scored based on a broad range of underlying indicators (e.g., the affordability of housing, public green spaced, public transport links, access to healthcare services, and the proportion of ethnic minority representation in the workplace) in a range of 0 to 1 where higher scores indicate more favourable conditions. A full list of underlying indicators used to calculate scores is available online [26].

The index was constructed using indicators from a variety of sources including national census data and land registry, data.police.UK, government petitions, and the National Health Service database. CWI data were compiled and matched to the #BeeWell 2021 dataset using the adolescents’ home postcodes (provided by GM local authorities). While total scores were used to quantify each domain to aid model convergence; the estimated mean scores for each indicator are also provided in Appendix A. The CWI segregates data across “Seamless Locales”, which served as the geographical unit used to discriminate neighbourhoods in the current study (see below). Mean neighbourhood scores for each CWI variable were treated as potential mediators of the relationship between neighbourhood deprivation and mental health trajectories.

#### 2.3.4. Seamless Locales and Neighbourhood Deprivation

*Seamless locales* are bespoke geographical units created by Geolytix that were designed to represent areas identifiable to individuals as their local neighbourhood [25]. In total, there are 28,317 seamless locales covering 100% of the UK, each containing an average of 2230 residents and 973 homes spanning 8.7 km^2^; 277 of these are represented in our sample of GM (Figure 1). In comparison to other commonly used geographic units, seamless locales are slightly less granular than Lower Super Output Areas (LSOAs)—the unit used for IMD rankings—but more granular than Middle Super Output Areas (MSOAs) or electoral wards.

IMD is the official measure of socio-economic deprivation in England that was developed by the UK Government’s Office for National Statistics. It was linked to the #BeeWell dataset using the participants’ postcodes. We calculated the mean IMD rank of each locale (freely available at the LSOA level and disaggregated to the individual-level according to participants’ home postcodes). To reduce response bias, we used data from every young person for whom we had home postcode data, regardless of whether they returned a #BeeWell survey (*N* = 98,244). This mean rank was then converted to a percentage and inverted such that a higher score on this continuous indicator reflected greater neighbourhood-level deprivation [27]. A heat map of GM is shown in Figure 1 that illustrates the derived IMD scores for each locale (Figure 1). Locale-specific CWI scores were also mapped and are provided in Appendix A. For those unfamiliar with the geography of GM, the city centre is located in the cluster of yellow locales toward the centre of the map, which indicates greater deprivation in the central neighbourhoods, as is typical of large cities.

#### 2.3.5. Covariates

Ethnicity and gender were included as covariates in the adjusted models due to their well-documented associations with adolescent mental health. Ethnicity was treated as a six-category variable (provided by parents to schools, which was then linked to the #BeeWell dataset) and distinguished between Asian, Black, Chinese, Mixed, White, and Any Other Ethnic Group. Dummy variables were derived with White used as the reference. Gender was treated as a two-category variable (sourced from school registers) and distinguished between boys and girls, with girls used as the reference.

It is important to note, in the UK context, minoritised ethnic groups are more likely to reside in disadvantaged or spatially segregated neighbourhoods due to historical and structural processes [28]. This collinearity often makes it challenging to disentangle the effects of neighbourhood from the sociodemographic composition of its inhabitants. To avoid masking meaningful contextual effects, we therefore present the results of the unadjusted model as the primary estimates in the main manuscript, with fully adjusted results that control for gender and ethnicity provided in Appendix A.

#### 2.3.6. Statistical Methods

A statistical analysis plan for the study was published on the Open Science Framework a priori [29]. All analyses were performed using Mplus, version 8.8, and the ‘*MplusAutomation*’ package in R [30,31]. Heat maps were generated using QGIS, version 3.34 [32]. Missing data for all variables were handled using Full Information Maximum Likelihood estimation (FIML) [33]. Analyses were conducted in two distinct phases: (1) identification of the classification model and (2) mediation analysis of the structural model. The following process was used to establish relationships with life satisfaction (model 1) and emotional difficulties (model 2) independently.

#### 2.3.7. Phase One: Identification of the Classification Model

A series of linear and non-linear latent growth mixture models (LGMMs) were enumerated to identify the class solution which fit the data best. In line with extant literature on latent class mediation [34], we adopted a maximum likelihood three-step approach whereby *k* + 1 class solutions were enumerated using a series of relative fit indices that are frequently used in latent variable modelling [35]. Elbow plots were generated to illustrate the point at which increased model complexity yielded diminished returns in model fit. Methodological researchers have urged caution on over-reliance on quantitative fit statistics, with current best-practice guidelines suggesting that equivalent emphasis should be placed on model interpretability [35]. Substantive criteria included sample distribution (where solutions identifying classes containing <1% of the sample were considered unstable), model parsimony (where the simplest solution was preferred), and the generalisability of the findings. This process was conducted prior to the inclusion of covariates so as to not bias the parameter estimates. Thereafter, posterior probabilities were used to fix class membership and avoid class switching when auxiliary variables were added to the model in phase two [36]. The data were clustered by school, with non-independence of observations controlled for using a sandwich estimator.

#### 2.3.8. Phase Two: Mediation Analysis of the Structural Model

In phase two, independent multi-level mediation models were constructed for life satisfaction (model 1) and emotional difficulties (model 2) where *x* is the neighbourhood deprivation (level 2), *m*^1^–*m*^9^ are the CWI items (level 2), and *y* is the trajectory class membership (level 1) using the best fitting latent growth mixture model identified in the previous step. Now that clustering of individual-level responses had been controlled for and class membership fixed, in phase two, the sandwich estimator was repurposed to account for clustering at the neighbourhood level.

To aid interpretation, the mediation model is graphically illustrated in Figure 2. A series of latent class regressions were run with *m^k^* regressed on *x* (α path), *y* regressed on *m^k^* (*b* path), and *y* regressed on *x* (c path) and on *x* and *m^k^* (c’ path). Mathematically speaking, c = c’ + (α*b*). Wald chi-square tests were used to estimate whether the product of pathways α and *b* was statistically different from zero for each trajectory class, with a significant result providing evidence of mediation. In each model, the largest class was used as the reference. All CWI items were added to the model concurrently to account for covariance between mediators.

## 3. Results

### 3.1. Phase One: Identification of the Classification Model

A full breakdown of the model fit statistics for the class enumeration phase of the analysis is provided in Appendix A, with a visual comparison of the BIC values graphically illustrated in Figure 3 and Figure 4. The within-class variance of indicator variables was held equal across classes in the first instance; however, convergence issues emerged for both models from the 4-class solution onward (solutions contained negative, non-significant slope factor residual variances); hence, the enumeration phase was repeated with within-class variance of slope factors for *k* + 1 models fixed at zero.

The fit statistics favoured non-linear solutions for both models, with the slope factor variances fixed at zero for life satisfaction and held equal for emotional difficulties. As is frequently the case in latent variable modelling with continuous indicators, the relative fit improved with each additional class. No clear and obvious elbow was visible to denote the point at which added model complexity led to diminished improvements in fit. However, a four-class (non-linear) model of life satisfaction and a three-class (non-linear) model of emotional difficulties offered a good balance of quantitative fit, sample distribution (i.e., no unstable classes), and model parsimony (i.e., a simple model that discriminated between distinct trajectories). Examination of the *k* + 1 model probability plots highlighted that these classes were retained in more complex iterations, suggesting that a stable solution was identified (Appendix A) [35].

Collectively, the fit indices and substantive criteria identified a four-class non-linear model for life satisfaction and a three-class non-linear model for emotional difficulties as the best fit to the data. Although classification entropy should not be treated as an indicator of model fit [37], our best fitting models were both above the minimum acceptable level of 0.60 [35] (life satisfaction: 0.66; emotional difficulties: 0.61). Probability plots of the final classification models are provided in Figure 5 and Figure 6.

#### 3.1.1. Life Satisfaction Model

The following life satisfaction trajectories from 2021 to 2023 were identified: a Consistently High class (n = 19,177; 71.0%; fairly stable scores of 7.855 to 7.375); a Deteriorating class (n = 1704; 6.3%; decreasing non-linearly from 7.873 to 2.161); an Improving class (n = 2358; 8.7%; increasing non-linearly from 2.979 to 7.485); and a Consistently Low class (n = 3770; 13.9%; stable scores of 3.780 to 3.974). Contrasting slightly from H^1a^, which predicted a large and consistently moderate life satisfaction trajectory, the sample was most heavily weighted toward the class with Consistently High life satisfaction (Figure 5).

#### 3.1.2. Emotional Difficulties Model

The following emotional difficulties trajectories from 2021 to 2023 were identified: a Low/Lessening emotional difficulties class (*n* = 14,211; 53.7%; scores decreasing non-linearly from 4.540 to 2.490); a Sub-Clinical class (*n* = 10,137; 38.3%; stable scores of 8.682 to 9.179); and an Elevated/Worsening class (*n* = 2114; 8.0%; scores increasing non-linearly from 10.924 to 16.708). In line with H^1b^, the sample was most heavily weighted toward the class with Low/Lessening emotional difficulties (Figure 6).

Although life satisfaction and emotional difficulties were modelled independently, a cross-tabulation indicating all possible combinations of class membership is provided in Table 3. To derive cross-model class proportions, the participants’ most likely class membership for each model was used and linked using the participant’s random ID. Note that although useful as an overview of, for example, the proportion of young people following Consistently High life satisfaction and Low/Lessening emotional difficulties trajectories simultaneously, class membership is probabilistic and actual cross-model proportions may differ slightly to those reported here.

The Odds Ratios and 95% Confidence Intervals of each life satisfaction and emotional difficulties trajectory (relative to any other combination) are provided in Table 4 to indicate the strength of the association between trajectories. The ORs were estimated by collapsing alternative trajectories into an “Other” category for each outcome, excluding missing values due to incomplete data, to create a series of 2 × 2 contingency tables, for instance, Consistently High and Low/Lessening (*n* = 13,148), Consistently High and Other emotional difficulties (*n* = 7195), Other life satisfaction and Low/Lessening (*n* = 1227), Other life satisfaction and Other emotional difficulties (*n* = 4170). The ORs were calculated as the cross-product ratio of these counts. Individuals in the Consistently High life satisfaction class had substantially increased odds of reporting *Low/Lessening* emotional difficulties and reduced odds of Sub-Clinical or Elevated/Worsening difficulties. The Consistently Low class showed the opposite pattern, with elevated odds of following problematic trajectories of emotional difficulties. The Improving and Deteriorating classes were associated with more mixed patterns. In particular, the weak/moderate association between *Deteriorating* life satisfaction Sub-Clinical emotional difficulties indicates that the constructs are related but do not overlap entirely. Although many associations reached statistical significance, caution is warranted as the large sample size increased the likelihood that even minimal effects achieved significance. Likewise, the estimates derived from low counts (e.g., in the Elevated/Worsening class) may be unstable and should also be interpreted cautiously.

#### 3.1.3. Phase Two: Multi-Level Mediation Analysis of the Structural Model

##### Total, Direct, and Indirect Effects of Neighbourhood Deprivation on Life Satisfaction Trajectories

Table 5 presents the total, direct, and indirect effects of neighbourhood deprivation (IMD) on life satisfaction trajectories, with the Consistently High life satisfaction trajectory serving as the reference category.

Total Effects (Before Inclusion of Mediators): Prior to accounting for the aspects of community wellbeing, neighbourhood deprivation demonstrated a significant total effect on two of the three life satisfaction trajectory classes, offering partial support for H^2b^. A higher neighbourhood deprivation was associated with increased odds of following Deteriorating (OR = 1.081, [1.023, 1.142]) and Consistently Low trajectories (OR = 1.084, [1.051, 1.119]). The total effect on the Improving trajectory class was not statistically significant (OR = 0.977, [0.936, 1.020]).

Direct Effects (After Adjustment for Mediators): After controlling for the aspects of community wellbeing, the effects of neighbourhood deprivation changed substantially. The direct effect of neighbourhood deprivation remained significant for both the Improving and the Consistently Low trajectory classes. A higher deprivation was associated with reduced odds of following an Improving (OR = 0.898 [0.832, 0.969]) trajectory (which starts low but improves over time) and increased odds of following a Consistently Low trajectory (OR = 1.133, [1.073, 1.197]). The direct effect on the Deteriorating trajectory class became non-significant (OR = 1.087, [0.987, 1.197]).

Indirect Effects Through Community Wellbeing: All α and *b* path coefficients are reported in Appendix A, respectively. Many of the α coefficients are statistically significant, providing support for investigating the indirect effects of deprivation through aspects community wellbeing. Of note, the α coefficients are almost identical for the life satisfaction and emotional difficulties models as the underlying samples were largely identical, excluding participants with missing information for either outcome variable. A significant positive indirect effect of Equality was observed for the Consistently Low trajectory class (α*b*^2^ = 0.016, [0.002, 0.029]), indicating that neighbourhood deprivation increases the risk of Consistently Low life satisfaction through increased equality. In other words, the indicators of equality in deprived neighbourhoods amplify the likelihood of poor life satisfaction outcomes. While contrary to H^3h^, we offer explanations for this seemingly unexpected finding in the discussion. A significant negative indirect effect of Housing, Space, and Environment was also identified for the Consistently Low trajectory class (α*b*^8^ = −0.026, [−0.046, −0.006]), in line with H^3e^. This negative indirect effect suggests that housing and environmental resources serve as a protective buffer, helping to mitigate the likelihood that neighbourhood deprivation leads to Consistently Low life satisfaction. The remaining aspects of community wellbeing—Education and Learning (H^3a^); Economy, Work, and Employment (H^3b^); Health (H^3c^); Culture, Heritage, and Leisure (H^3d^); Transport, Mobility, and Connectivity (H^3f^); Relationships and Trust (H^3g^); and Voice and Participation (H^3i^)—did not demonstrate significant indirect effects for any of the life satisfaction trajectory classes.

##### Total, Direct, and Indirect Effects of Neighbourhood Deprivation on Emotional Difficulties Trajectories

Table 5 presents the corresponding analysis for the emotional difficulties trajectory classes, with the reference category being Low/Lessening emotional difficulties.

Total Effects (Before Inclusion of Mediators): Prior to accounting for the aspects of community wellbeing, greater neighbourhood deprivation reduced the odds of following a Sub-Clinical trajectory (OR = 0.975, [0.954, 0.996]). The total effect on the *Elevated/Worsening* symptoms trajectory was not statistically significant (OR = 0.0996, [0.954, 1.040]); hence, the findings offer partial support for H^2c^.

Direct Effects (After Adjustment for Mediators): After controlling for the aspects of community wellbeing, the direct effects of neighbourhood deprivation on emotional difficulties trajectories became non-significant for both Sub-Clinical (OR = 0.992, [0.952, 1.033]) and Elevated/Worsening symptoms trajectories (OR = 1.060, [0.985, 1.141]). This attenuation suggests that the collective effect of all aspects of community wellbeing fully explain the relationship between deprivation and emotional difficulties in young people.

Indirect Effects Through Community Wellbeing: All α and *b* path coefficients are reported in Appendix A, respectively. A significant negative indirect effect of Housing, Space, and Environment was identified for the Sub-Clinical trajectory (α*b*⁸ = −0.026, [−0.045, −0.008]), in line with H^4e^. This negative indirect effect indicates that better housing and environmental conditions help to buffer the development of Sub-Clinical emotional difficulties in the context of neighbourhood deprivation. All other aspects of community wellbeing—Education and Learning (H^4a^); Economy, Work, and Employment (H^4b^); Health (H^4c^); Culture, Heritage, and Leisure (H^4d^); Transport, Mobility, and Connectivity (H^4f^); Relationships and Trust (H^4g^); Equality (H^4h^); and Voice and Participation (H^4i^)—did not demonstrate significant indirect effects for either emotional difficulties trajectory class.

##### Adjusting for Covariates

The total and direct effects of neighbourhood deprivation in the fully adjusted models that additionally controlled for ethnicity and gender largely coincided with those of the primary, unadjusted models (Appendix A). However, as expected, a key divergence emerged in the mediation pathways: after inclusion of gender and ethnicity, none of the indirect effects via the hypothesised mediators reached statistical significance. This attenuation suggests that the mechanisms linking neighbourhood exposures to adolescent wellbeing may be closely intertwined with sociodemographic context, particularly with respect to structural patterns of residential stratification and differential access to social and environmental resources. In other words, adjusting for gender and ethnicity absorbed the variance that is inherently part of the pathway through which the neighbourhood context exerts its influence, highlighting the complexity of disentangling compositional and contextual effects in observational research.

## 4. Discussion

The academic literature is largely in agreement that adolescent life satisfaction declines over time [38], with the overall rates dropping substantially in recent years [39]. Our findings offer a different perspective, demonstrating that the decline is not as universal as alternative reports and methodological approaches might suggest [38]. Alongside Improving, Deteriorating, and Consistently Low trajectories, the majority of adolescents studied herein were observed as following Consistently High levels of life satisfaction, defined by the Office for National Statistics as scores ≥ 7, over two years [21]. This could be evidence that a longer survey period is required to observe the drastic changes described elsewhere, or a positive sign that mental health service provision in GM is having a beneficial effect. Nonetheless, a significant minority of adolescents followed a concerningly poor and unchanging trajectory, implying that service provision may be inconsistent or insufficiently capturing those at greatest risk [40]. Persistently low life satisfaction increases vulnerability to emotional difficulties during the transition to adulthood and may exacerbate socioeconomic and health inequalities [41]. Given the rising demand for mental health services, a greater emphasis on proactive, upstream interventions—such as school-based wellbeing programmes, community engagement initiatives, and targeted support for at-risk youth—could help alleviate pressures on crisis services and reduce the long-term societal burden of sub-optimal life satisfaction.

Enhancing life satisfaction during adolescence yields numerous potential benefits, including improved academic performance, higher earning potential, and better social and physical health outcomes [42,43]. From an economic perspective, even for individuals following a Consistently High trajectory, efforts to improve life satisfaction may have significant societal value. Despite mean scores remaining above the ‘high’ threshold, the marginal decline observed in this class carries an estimated annual per capita cost of GBP 1560.00 (GBP 1930.00 from Year 8 to 9; GBP 1189.00 from Year 9 to 10). The economic burden is substantially greater for the Deteriorating class, with the rate of change having an estimated cost of GBP 22,951.00 from Year 8 to 9, and GBP 14,176.50 from Year 9 to 10 (GBP 18,564.00 on average). These findings provide strong justification for investment in cost-effective public health interventions, particularly those estimated to cost less than the projected economic loss associated with trajectories of decline observed herein. Notably, more pronounced changes—both improvements and decline—occurred during the transition from Year 8 to Year 9, suggesting that this period may serve as a critical window for intervention.

Cross-classification information on life satisfaction and emotional difficulties class membership (Table 3 and Table 4) provides useful insight into the alignment—and divergence—of adolescent mental health experiences across affective and evaluative domains. The findings demonstrated both expected and non-intuitive patterns of co-occurrence, reinforcing the importance of dual-state approaches over traditional unidimensional models that aggregate positively and negatively oriented domains into a single, composite indicator of mental health. The most common trajectory pairing was the most desirable scenario—Consistently High life satisfaction with Low/Lessening emotional difficulties—which, according to most likely class estimation, was experienced by 13,148 adolescents (47.41%). This offers a tentative indication of the proportion of conceivable candidates for flourishing. This estimate is broadly consistent with previously reported prevalence rates both in the UK (between 18 and 59%,depending on evaluations of family connectedness) [44] and internationally (46% in Indonesia [45], 46% in India [46], 42% in Australia [47], and 40% in the United States [48]).

However, a substantial proportion also followed trajectories indicative of problematic life satisfaction and/or emotional difficulties, underscoring the pressing need to enhance early identification and intervention strategies aimed at supporting mental health during this critical developmental period. There is evidence that suggests that many of those with Sub-Clinical symptoms are at heightened risk of developing clinically relevant symptoms in the near future [49]. Individuals following this trajectory of symptoms therefore present a major challenge for public health systems, particularly in the context of already overburdened services.

In England, Child and Adolescent Mental Health Services (CAMHS) are under increasing strain, with a 50% rise in referrals since 2019 [50]. In response, recent policy recommendations for GM have highlighted several key priorities for improving mental health crisis management services. These include greater clarity around available services, enhanced integration between service providers, and the expansion of crisis services to cover 16–18 year olds [40]. Addressing these systematic challenges is an essential next step in resolving the ongoing adolescent mental health epidemic.

### 4.1. Neighbourhood Deprivation

The findings demonstrate that neighbourhood deprivation is a salient predictor of longitudinal patterns of adolescent life satisfaction (Table 5). Specifically, the estimated total effects indicate that those residing in more deprived neighbourhoods are significantly more likely to follow Deteriorating and Consistently Low life satisfaction trajectories than a Consistently High class. These results are consistent with a substantial body of literature suggesting that structural features of neighbourhoods—such as economic disadvantages, limited access to green space, and underinvestment in local services—are associated with reduced adolescent wellbeing [51]. The increased odds of following problematic life satisfaction trajectories in the context of higher deprivation also supports the view that such environments may erode protective factors that typically support favourable life satisfaction, including opportunities for social participation, safety, and a sense of control over one’s future [7].

Although no significant association was observed for membership in the Elevated/Worsening emotional difficulties class, greater neighbourhood deprivation was paradoxically associated with slightly lower odds of following a Sub-Clinical emotional difficulties trajectory (Table 6). This pattern warrants careful interpretation. Prior research has suggested that structural deprivation may exert more immediate influence on life satisfaction, while emotional symptoms may be more strongly shaped by proximal psychosocial influences [13]. Divergent results across outcome domains and the absence of parallel effects between neighbourhood deprivation, life satisfaction, and emotional difficulties strengthens our understanding in this regard. Specifically, they underscore the multidimensional nature of adolescent mental health, indicating different sensitivities of internalising versus cognitive–evaluative outcomes to environmental stressors. Interventions to enhance adolescent wellbeing in deprived neighbourhoods may require differentiated approaches: universal strategies to support evaluative wellbeing may be sufficient, but more targeted efforts may be necessary to address emergent emotional difficulties.

### 4.2. Indirect Effects Through Aspects of Community Wellbeing

The mediation analyses identified the mechanisms through which neighbourhood effects operate, highlighting the complexity and nuance in how a young person’s neighbourhood may shape both the structural conditions of daily life and subsequent mental health trajectories. Better Housing, Space, and Environment most consistently supported dual-state mental health, reducing the odds of Consistently Low life satisfaction and Sub-Clinical emotional difficulties for adolescents in deprived neighbourhoods.

#### 4.2.1. Housing, Space, and Environment

Affordable housing is a key feature of Mental-Health-Friendly Cities [52]. Likewise, access to green space is widely accepted as being good for mental health. Natural England’s standard measure of neighbourhood green space reported that 53% of UK households have access to green space within a realistically walkable distance [53]; however, this figure masks the significant heterogeneity between urban areas. GM’s industrial heritage has shaped its urban landscape, leading to high-density housing in the central areas (as shown in Appendix A). Consequently, there is limited access to parks and green space near the city centre but an abundance of such areas on the periphery. This geographic distribution is important. In Manchester, as is the case in many cities globally, many deprived neighbourhoods are concentrated in or near the city centre (Figure 1). As a result, adolescents in these neighbourhoods must often contend with the impact of socio-economic hardship (e.g., lack of food security and housing instability) while having fewer natural mechanisms of support available than residents of more affluent suburban areas.

That said, the interconnectedness and collective effect of the myriad of factors that comprise Housing, Space, and Environment remain unclear. Indeed, research in this cohort, as well as in other studies, has reported null associations between green space and both life satisfaction and internalising symptoms longitudinally [7,54]. Additionally, evidence for the impact of other factors related to Housing, Space, and Environment (such as overcrowding, air pollution, and traffic) is limited for this age group, with further research needed to determine whether factors beyond housing affordability should be prioritised in local public health policy.

#### 4.2.2. Equality

The positive association between Equality and Consistently Low life satisfaction may appear counter-intuitive initially but can be better understood by examining the indicators contributing to the CWI Equality score: House Price Gap, Second Home Ownership, Proximity to Independent Schools, Qualification Gap, Ethnic Minority Representation in Professional Occupations, Income Inequality, and Long-term Housing Security. In more deprived areas, social housing is common, with little variation in value, and second home ownership is rare. Similarly, proximity to independent schools is less relevant in predominantly state-funded education systems. As a result, greater ‘equality’ in these neighbourhoods likely reflects widespread socio-economic disadvantages rather than genuine parity with more advantaged areas across a city region. Deprived neighbourhoods are also inhabited by a higher proportion of minoritised ethnic groups, naturally leading to greater representation in the workforce further enhancing proposed equality in disadvantaged areas [55].

### 4.3. Study Limitations

While we propose that adolescents following trajectories characterised by Consistently High life satisfaction and Low/Lessening emotional difficulties may be plausible candidates for what might be termed “flourishing” [12], a more comprehensive conceptualisation is warranted to adequately capture its prevalence. Flourishing is inherently multidimensional, and although there is no universally accepted definition, it is broadly understood to encompass more than emotional wellbeing alone. Indeed, to be truly flourishing—insofar as such a state is possible, given that flourishing is an asymptotic ideal that can be approached but never fully attained—young people must also report favourable perceptions across multiple domains that were not measured herein, including meaning and purpose, physical health, social relationships, volitional autonomy, and financial stability [56,57]. Future research should seek to incorporate multidimensional indicators of flourishing across time in order to examine whether distinct developmental trajectories emerge when these broader facets of wellbeing are considered.

Although this longitudinal research adds value to a field dominated by cross-sectional studies, the tracking of young people across three timepoints spanning two years may be insufficient to fully capture the extent to which positively and negatively orientated domains of mental health vary throughout adolescence. Indeed, the examination of sample-level mean scores (Appendix A) revealed little to no change across the study period. While this adds further justification for the use of mixture modelling, which we contend captures a large portion of the heterogeneity across this period, researchers are likely to find more pronounced shifts with an extended period of observation. Of particular interest would be studies that track changes in life satisfaction and emotional difficulties throughout the transition from adolescence to early adulthood to establish the extent to which neighbourhood environments in youth exert sustained influence on mental health outcomes in future life stages.

Our continuous measure of neighbourhood deprivation was derived from the mean Index of Multiple Deprivation (IMD) scores across all adolescents within each locale for whom we had postcode data. To mitigate the risk of reporter bias and enhance representativeness, we used postcode and IMD data from over 98,000 adolescents—far exceeding the number responding to the #BeeWell survey alone—who may disproportionately represent less deprived areas. The median number of pupils contributing to each locale-specific deprivation score was 261, which provides a strong support for the reliability of these estimates. As shown in Figure 1, this approach produced deprivation scores that align closely with the expected regional variation. Nevertheless, in locales with relatively small sample sizes, the precision of the derived scores may be reduced due to greater sampling variability.

## 5. Conclusions

This study identified heterogeneous trajectories of adolescent mental health, providing empirical support for all three research questions. While the most common patterns—Consistently High life satisfaction and Low/Lessening emotional difficulties—indicated general stability (supporting H^1a–b^), a substantial minority followed concerning and problematic trajectories, including Consistently Low or Deteriorating life satisfaction and Sub-Clinical emotional difficulties trajectories. The evidence presented herein reinforces the need for early, sustained interventions and highlights the importance of disaggregating affective and evaluative domains in adolescent mental health research.

Neighbourhood deprivation emerged as a consistent predictor of problematic life satisfaction outcomes, supporting H^2a^ and H^2b^. However, deprivation was not associated with Elevated/Worsening emotional difficulties, and was unexpectedly linked to lower odds of Sub-Clinical difficulties. This suggests that life satisfaction may be more sensitive to macro-level structural stressors, whereas emotional symptoms could emerge in response to more proximal or relational factors. The apparent protective effect may also reflect social cohesion in disadvantaged neighbourhoods or differential recognition of distress symptoms.

The mediation analyses (H^3^) partially explained these associations. The Housing, Space, and Environment aspect of community wellbeing emerged as a key protective factor for both life satisfaction and emotional difficulties, particularly in deprived neighbourhoods. Conversely, positive associations between neighbourhood ‘Equality’ and Consistently Low life satisfaction likely reflect artefacts of deprivation, rather than genuine parity.

Overall, the findings point to the need for multidimensional, context-sensitive mental health policy incorporating structural, environmental, and psychosocial components. Adolescence—particularly the transition from Year 8 to Year 9—may represent a critical window for intervention. Prioritising affordable housing, environmental improvements, and accessible community support schemes could mitigate emerging inequalities and enhance trajectories of mental health for adolescents.

## Figures and Tables

**Figure 1 ijerph-22-00951-f001:**
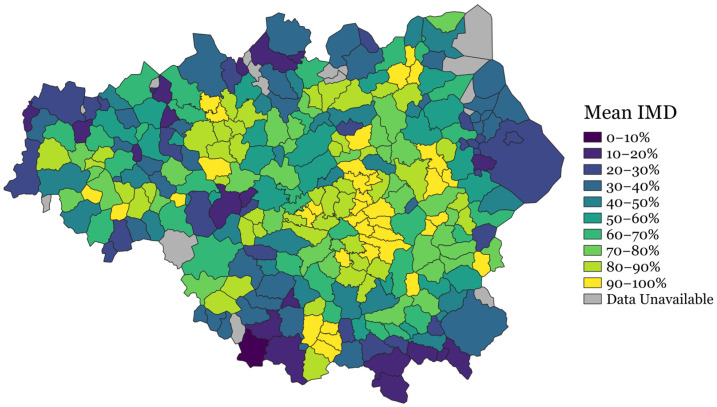
Heat map of mean IMD scores attributed to each GM locale with higher scores reflecting greater deprivation.

**Figure 2 ijerph-22-00951-f002:**
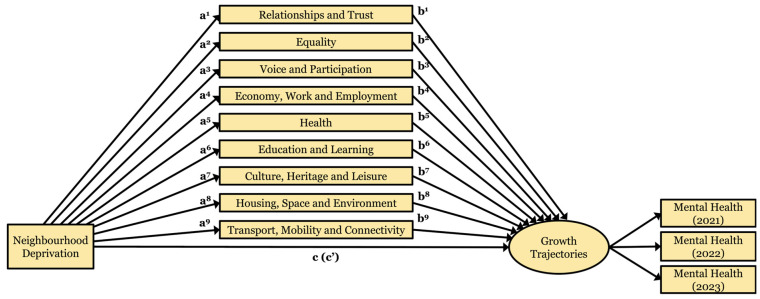
Mediation model to test neighbourhood effects on dual-state mental health trajectories in adolescence.

**Figure 3 ijerph-22-00951-f003:**
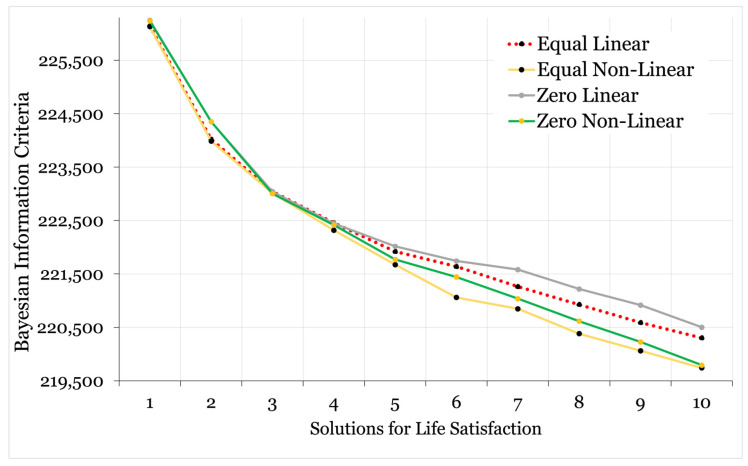
Elbow plot illustrating linear and non-linear growth models with variances freely estimated and held equal across groups or fixed at zero (life satisfaction).

**Figure 4 ijerph-22-00951-f004:**
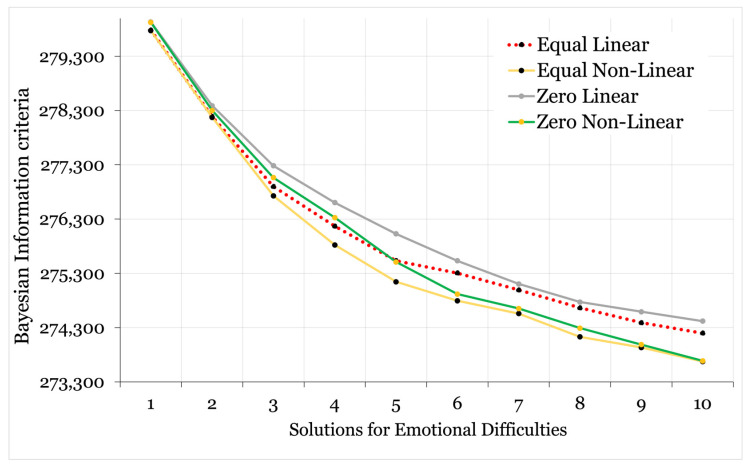
Elbow plot illustrating linear and non-linear growth models with variances freely estimated and held equal across groups or fixed at zero (emotional difficulties).

**Figure 5 ijerph-22-00951-f005:**
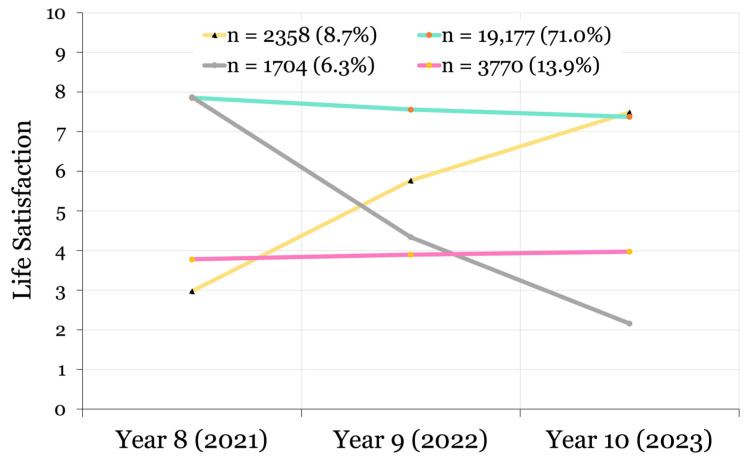
Latent growth curves identified in the best fitting model demonstrating trajectories of life satisfaction in adolescents from 2021 to 2023.

**Figure 6 ijerph-22-00951-f006:**
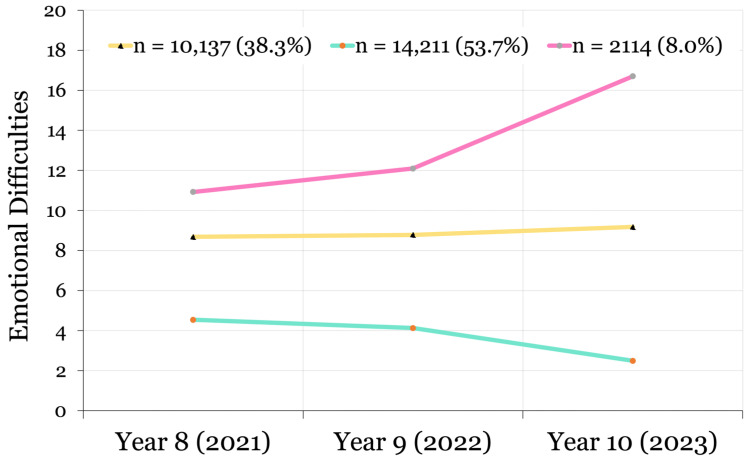
Latent growth curves identified in the best fitting model demonstrating trajectories of emotional difficulties in adolescents from 2021 to 2023.

**Table 1 ijerph-22-00951-t001:** Hypotheses and the aspect of community wellbeing they relate to.

	Hypothesis	
Aspect of Community Wellbeing	Life Satisfaction	Emotional Difficulties
Education and Learning	H^3a^	H^4a^
Economy, Work, and Employment	H^3b^	H^4b^
Health	H^3c^	H^4c^
Culture, Heritage, and Leisure	H^3d^	H^4d^
Housing, Space, and Environment	H^3e^	H^4e^
Transport, Mobility, and Connectivity	H^3f^	H^4f^
Relationships and Trust	H^3g^	H^4g^
Equality	H^3h^	H^4h^
Voice and Participation	H^3i^	H^4i^

**Table 2 ijerph-22-00951-t002:** Demographic information and descriptive statistics for the analytical samples.

			Life Satisfaction Model		Emotional Difficulties Model	
Variable		Metric	Value	Missing (%)	Value	Missing (%)
Sample Size		N	27,009	-	26,461	-
Schools		N	185	0.00	179	0.00
Girls		n (%)	13,477 (50.10)	0.42	13,226 (50.20)	0.40
SEN		n (%)	4393 (16.70)	2.83	4147 (16.10)	2.70
FSM		n (%)	7283 (27.90)	3.43	7122 (27.80)	3.34
EAL		n (%)	5437 (20.70)	2.68	5326 (20.70)	2.57
IMD		Mean (sd)	66.90 (22.91)	5.49	66.88 (22.92)	5.41
Ethnicity	White	n (%)	17,217 (66.60)	4.23	16,927 (66.70)	4.14
	Any Other Ethnic Group	n (%)	658 (2.50)	4.23	639 (2.50)	4.14
	Asian	n (%)	4410 (17.00)	4.23	4310 (17.00)	4.14
	Black	n (%)	1655 (6.40)	4.23	1612 (6.40)	4.14
	Chinese	n (%)	258 (1.00)	4.23	255 (1.00)	4.14
	Mixed	n (%)	1669 (6.50)	4.23	1623 (6.40)	4.14
CWI	Relationships and Trust	Mean (sd)	0.69 (0.06)	5.41	0.70 (0.06)	5.32
	Equality	Mean (sd)	0.54 (0.11)	5.41	0.54 (0.11)	5.32
	Voice and Participation	Mean (sd)	0.55 (0.08)	5.41	0.55 (0.08)	5.32
	Economy, Work, and Employment	Mean (sd)	0.66 (0.09)	5.41	0.66 (0.09)	5.32
	Health	Mean (sd)	0.53 (0.13)	5.41	0.53 (0.13)	5.32
	Education and Learning	Mean (sd)	0.87 (0.07)	5.41	0.87 (0.07)	5.32
	Culture, Heritage, and Leisure	Mean (sd)	0.71 (0.10)	5.41	0.71 (0.10)	5.32
	Housing, Space, and Environment	Mean (sd)	0.51 (0.09)	5.41	0.51 (0.09)	5.32
	Transport, Mobility, and Connectivity	Mean (sd)	0.76 (0.06)	5.41	0.76 (0.06)	5.32
Latent Class Indicator	Total Score in 2021	Mean (sd)	6.86 (2.48)	32.86	6.60 (4.62)	33.48
	Total Score in 2022	Mean (sd)	6.63 (2.47)	39.49	6.58 (4.88)	40.60
	Total Score in 2023	Mean (sd)	6.62 (2.43)	42.23	6.26 (4.95)	43.11

SEN, have special educational needs; FSM, eligible for free school meals; EAL, speak English as an additional language; IMD, Index of Multiple Deprivation; CWI, Community Wellbeing Index.

**Table 3 ijerph-22-00951-t003:** Cross-tabulation of all possible combinations of mental health trajectories and their proportions.

		Emotional Difficulties, *n* (%)			
		Low/Lessening	Sub-Clinical	Elevated/Worsening	Missing
Life Satisfaction *n* (%)	Consistently High	13,148 (47.41%)	6823 (24.61%)	372 (1.34%)	1017 (3.67%)
	Improving	478 (1.72%)	662 (2.39%)	44 (0.16%)	37 (0.13%)
	Deteriorating	201 (0.73%)	391 (1.41%)	274 (0.99%)	20 (0.07%)
	Consistently Low	548 (1.98%)	2330 (8.40%)	469 (1.69%)	195 (0.70%)
	Missing	461 (1.66%)	247 (0.89%)	13 (0.05%)	NA

**Table 4 ijerph-22-00951-t004:** Odds Ratios and 95% Confidence Intervals for the association between life satisfaction (rows) and emotional difficulties (columns) trajectory class membership.

	Emotional Difficulties Class (vs. All Others)		
Life Satisfaction Class (vs. All Others)	Low/Lessening	Sub-Clinical	Elevated/Worsening
Consistently High	6.212 * [5.790, 6.657]	0.301 * [0.283, 0.320]	0.109 * [0.096, 0.124]
Improving	0.519 * [0.461, 0.584]	1.994 * [1.773, 2.242]	0.811 [0.597, 1.103]
Deteriorating	0.228 * [0.194, 0.267]	1.263 * [1.101, 1.447]	12.549 * [1.718, 14.700]
Consistently Low	0.121 * [0.110, 0.133]	4.222 * [3.896, 4.565]	5.128 * [4.533, 5.797]

* Statistically significant.

**Table 5 ijerph-22-00951-t005:** Total, direct, and indirect effects of neighbourhood deprivation and community wellbeing on life satisfaction trajectories.

		Life Satisfaction Trajectory ^a^		
Pathway	Mediator	Improving	Deteriorating	Low
*Total effect of nb-IMD (before inclusion of mediators)*				
c OR [95% CI]		0.977 [0.936, 1.020]	1.081 [1.023, 1.142] *	1.084 [1.051, 1.119] *
*Direct effect of nb-IMD (adjusted for mediators)*				
c’ OR [95% CI]		0.898 [0.832, 0.969] *	1.087 [0.987, 1.197]	1.133 [1.073, 1.197] *
*Indirect (mediated) effect (product of paths a* × *b)*				
α*b*^1^ [95% CI] ^b^	*Rel. and Trust*	0.000 [−0.004, 0.005]	−0.001 [−0.007, 0.004]	0.000 [−0.003, 0.003]
α*b*^2^ [95% CI] ^b^	*Equality*	−0.007 [−0.025, 0.010]	0.004 [−0.013, 0.021]	0.016 [0.002, 0.029] *
α*b*^3^ [95% CI] ^b^	*Voice and Part.*	0.003 [−0.003, 0.009]	0.000 [−0.008, 0.008]	−0.002 [−0.007, 0.002]
α*b*^4^ [95% CI] ^b^	*Ec., Work, and Empl.*	0.051 [−0.015, 0.117]	−0.014 [−0.081, 0.052]	−0.014 [−0.053, 0.025]
α*b*^5^ [95% CI] ^b^	*Health*	0.000 [−0.019, 0.019]	−0.007 [−0.023, 0.010]	0.002 [−0.010, 0.013]
α*b*^6^ [95% CI] ^b^	*Edu. and Learn.*	−0.005 [−0.018, 0.009]	0.011 [−0.006, 0.027]	−0.007 [−0.016, 0.003]
α*b*^7^ [95% CI] ^b^	*Cult., Heri., and Leis.*	−0.001 [−0.005, 0.003]	0.001 [−0.004, 0.005]	0.002 [−0.006, 0.009]
α*b*^8^ [95% CI] ^b^	*Hous., Spac., and Env.*	0.019 [−0.014, 0.051]	0.026 [−0.008, 0.060]	−0.026 [−0.046, −0.006] *
α*b*^9^ [95% CI] ^b^	*Trans., Mob., and Conn.*	0.005 [−0.020, 0.029]	−0.028 [−0.060, 0.004]	−0.006 [−0.023, 0.010]

nb-IMD: Neighbourhood Mean Index of Multiple Deprivation Scores. * Statistically significant; ^a^ reference class: Consistently High life satisfaction; ^b^ differences between parameters were tested using the MODEL CONSTRAINT command in Mplus. Wald tests were used to evaluate whether the product of the predictor-to-mediator and mediator-to-outcome paths (i.e., the indirect effect) was significantly different from zero within each class. Estimates represent unstandardised indirect effects.

**Table 6 ijerph-22-00951-t006:** Total, direct, and indirect effects of neighbourhood deprivation and community wellbeing on emotional difficulties trajectories.

		Emotional Difficulties Trajectory ^a^	
Pathway	Mediator	Sub-Clinical	Worsening
*Total effect of nb-IMD (before inclusion of mediators)*			
c OR [95% CI]		0.975 [0.954, 0.996] *	0.996 [0.954, 1.040]
*Direct effect of nb-IMD (adjusted for mediators)*			
c’ OR [95% CI]		0.992 [0.952, 1.033]	1.060 [0.985, 1.141]
*Indirect (mediated) effect (product of paths a*b)*			
α*b*^1^ [95% CI] ^b^	*Rel. and Trust*	0.000 [−0.003, 0.002]	−0.002 [−0.005, 0.002]
α*b*^2^ [95% CI] ^b^	*Equality*	0.005 [−0.003, 0.014]	0.007 [−0.007, 0.020]
α*b*^3^ [95% CI] ^b^	*Voice and Part.*	−0.004 [−0.009, 0.001]	0.001 [−0.006, 0.007]
α*b*^4^ [95% CI] ^b^	*Ec., Work, and Empl.*	0.007 [−0.025, 0.040]	−0.033 [−0.083, 0.017]
α*b*^5^ [95% CI] ^b^	*Health*	0.006 [−0.002, 0.014]	−0.007 [−0.023, 0.009]
α*b*^6^ [95% CI] ^b^	*Edu. and Learn.*	−0.002 [−0.009, 0.005]	−0.008 [−0.020, 0.004]
α*b*^7^ [95% CI] ^b^	*Cult., Heri., and Leis.*	0.000 [−0.001, 0.001]	0.002 [−0.005, 0.008]
α*b*^8^ [95% CI] ^b^	*Hous., Spac., and Env.*	−0.026 [−0.045, −0.008] *	−0.017 [−0.042, 0.008]
α*b*^9^ [95% CI] ^b^	*Trans., Mob., and Conn.*	0.010 [−0.004, 0.025]	0.007 [−0.015, 0.029]

nb-IMD: Neighbourhood Mean Index of Multiple Deprivation Scores. * Statistically significant; ^a^ reference class: Low/Lessening emotional difficulties; ^b^ differences between parameters were tested using the MODEL CONSTRAINT command in Mplus. Wald tests evaluated whether the product of the predictor-to-mediator and mediator-to-outcome paths (i.e., the indirect effect) was significantly different from zero within each class. Estimates represent unstandardised indirect effects.

## Data Availability

An anonymised version of the #BeeWell survey responses will be made publicly available in 2026. Due to ethical governance constraints, this cannot be brought forward since the participants have been given the right to withdraw their data until this point, necessitating the need to maintain a securely stored pseudonymised version until 2026. In addition, linked administrative data (e.g., sex and free school meal eligibility) will never be shared publicly due to the prohibition of onward sharing in the data-sharing agreement in place with the Local Authorities who provided it. The Mplus syntax used to analyse the data can be made publicly available via the Open Science Framework upon acceptance of this manuscript for publication.

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
