# Peer review of "Local Landscapes, Evolving Minds: Mechanisms of Neighbourhood Influence on Dual-State Mental Health Trajectories in Adolescence"

_ijerph, 2025, doi:10.3390/ijerph22060951_

Round 1

Reviewer 1 Report

Comments and Suggestions for Authors

Review:  Local Landscapes, Evolving Minds:  Mechanisms of Neighbourhood influence on Dual-State Mental Health Trajectories in Adolescence

This manuscript takes advantage of a substantially large metropolitan panel survey of adolescents (aged 12-15)  in an attempt to identify classes of trajectories (over a three periods) for both positive mental health status (life satisfaction) and for negative mental health status (emotional difficulties) and to relate a set of contextual/neighborhood factors on the location of individuals in these derived classes.    A key aim is to examine how neighbourhood deprivation works through (mediated) other measures of neighbourhood well-being to influence the trajectory class of mental health (positive or negative) that the adolescent belongs to in the study.  Results/conclusions about the import of neighbourhood quality on classes are discussed; the authors suggest some qualities may buffer the effect/association between deprivation and poor classes of trajectories.  There are positives in the overall orientation of the work but some features temper my assessment.

Overall the manuscript is well-written.  It does need some clarification on certain points (see fixes/clarifications below)

Below are some points that suggest additional analyses are necessary.  In addition, some features of the data and measures require clarification.

  • Nature of Dual-State Mental Health

The authors appropriately make a point of the value in looking at both positive and negative mental health (MH from here out) and for the value the person-centered nature of their class trajectories.  But the analyses basically remain siloed as an analysis of positive MH and negative MH.   Nowhere do we see the overlap (association) between the classes of positive MH and negative MH.  What is the cell size for being in “improving satisfaction class” and the “low emotional difficulties” class.  This might inform why one sees or does not see some similarity in the separate analyses of what drives belonging to 1 of the 4 life satisfaction vs what drives belonging to 1 of the 3 emotional difficulties classes.  Another take on this is if you are concerned about the value of person centered analysis why not build the classes out of joint trajectories; it would be easy to model classes based on patterns of growth in satisfaction AND in difficulties.  This would seem to capture the dual nature of their discussion.  But minimally we have no sense of the overlap of these current categories.

  • Clarity in the mediation model

Modeling mediation requires having a reasonable causal framework.  While it is not fully clear what has gone into the deprivation score (see comment below) it still has to be argued that deprivation drives economy/work/employment in their model and that deprivation is not a result of economy/work/employment … there are other reverses or reciprocal specifications that could also be pointed out (e.g. housing/space/environment).  This makes it difficult, in general, to understand the direction of the association from deprivation to neighborhood well-being measures and the decomposition of the direct and indirect effects of deprivation.   The mediation concept is stretched here.  There may be more value (and consistent with their logic for the dual-state) by having just two latent constructs predicting MH classes.  One being deprivation oriented (negative neighbourhood quality) and one being resilient oriented (positive neighbourhood quality).   This would also line up with more recent literature that argues that urban context not just be captured as disadvantaged but also by positive features (advantaged/resilient).

  • The bias in the mediation model

In section 2.4.2 the authors explain they run the mediation model one intervening variable at a time.  They cite an issue of minimizing Type II error but each of these assessed “b” paths in their figure is highly biased by a set of left out variables.  The magnitude of these effects will be highly related to the inter-correlations among the one variable of focus and the left out remaining variables.  The c’ effect will likewise be biased and have differential size depending on deprivation’s differential relations to each of the well-being factors.  The only way this is not having an impact is if the correlations among the 9 factors of well-being are reasonably the same which is doubtful.   But as it is their concern for Type II error is, in my opinion, rather trivial compared to the bias and how that bias may drive differences in the parameters of high interest to them.   (I believe they would be better off just showing the result of their sensitivity analysis in detail).  

These two concerns (2 and 3) generate a level of uncertainty about the results that make it difficult to evaluate the conclusions and discussion of what matters to the two outcomes (positive or negative MH).

  • Other meaningful covariates.

It is also difficult to assess some of the estimates without including some rather typical covariates in the models.  Such things as gender(sex) or race/ethnicity are associate are rather basic to the assessment of MH.  And such things as race/ethnicity might also be related to neighborhood location (a role of economic/social segregation).  If the individual characteristic is not controlled some of its effects/association may be captured by neighbourhood.  Again, leading to potentially biased assessments of these key elements of interest.  

Fixes/Needing Clarification:

  • The authors acknowledge a clustering by school but there seems little concern for geographical clustering or the hierarchical nature of the youth nested within the neighborhood.

  • Table 3 and 4 could largely be removed and with the results of their choice of which class structure to believe in summarized and using Figure 3.

  • It is difficult to understand where the values for the neighborhood are coming from and their exact nature. This is especially the case for the IMD/deprivation index.  It is not clear what goes into it.  Is it an aggregate measure derived from the youth or from some other source.   If  both the well-being and deprivation are summed qualities provided by the youth how accurate can their assessment of such things as employment or housing or equality be?   Bottomline:  it is very unclear where these neighborhood assessments are coming from… maybe the authors are assuming a greater familiarity with these ecological data than the typical reader.

  • Table 7 is exceedingly dense. I would maybe remove the detailed variable label (table note them).  Or perhaps have one table for Satisfaction and one for Difficulties.

  • Also in table 7 … the “Chi-Square” [95% CI] is rather left unexplained. I’d prefer seeing the estimated coefficient of a*b … 

  • I think it is better to label Phase I as identifying the classification model (see section 2.4.1).

Citations…considerations.

  • Seems difficult to read a piece on neighborhood effects that doesn’t acknowledge the work out of Chicago’s neighborhood project by Robert Sampson, among others.  Especially the different ways of measuring neighborhood factors and also approaches to testing.  An early review… “Assessing Neighborhood Effects:  Social Processes and New Directions in Research”  Annual Review of Sociology 2002(Vol 28) …Sampson, Morenoff, and Gannon-Rowley.  But there are many others.

Author Response

This manuscript takes advantage of a substantially large metropolitan panel survey of adolescents (aged 12-15) in an attempt to identify classes of trajectories (over a three periods) for both positive mental health status (life satisfaction) and for negative mental health status (emotional difficulties) and to relate a set of contextual/neighborhood factors on the location of individuals in these derived classes. A key aim is to examine how neighbourhood deprivation works through (mediated) other measures of neighbourhood well-being to influence the trajectory class of mental health (positive or negative) that the adolescent belongs to in the study. Results/conclusions about the import of neighbourhood quality on classes are discussed; the authors suggest some qualities may buffer the effect/association between deprivation and poor classes of trajectories.  There are positives in the overall orientation of the work but some features temper my assessment. Overall, the manuscript is well-written. It does need some clarification on certain points (see fixes/clarifications below)

Below are some points that suggest additional analyses are necessary. In addition, some features of the data and measures require clarification.

  1. Nature of Dual-State Mental Health

The authors appropriately make a point of the value in looking at both positive and negative mental health (MH from here out) and for the value the person-centered nature of their class trajectories. But the analyses basically remain siloed as an analysis of positive MH and negative MH. Nowhere do we see the overlap (association) between the classes of positive MH and negative MH. What is the cell size for being in “improving satisfaction class” and the “low emotional difficulties” class.  This might inform why one sees or does not see some similarity in the separate analyses of what drives belonging to 1 of the 4 life satisfaction vs what drives belonging to 1 of the 3 emotional difficulties classes. Another take on this is if you are concerned about the value of person centered analysis why not build the classes out of joint trajectories; it would be easy to model classes based on patterns of growth in satisfaction AND in difficulties. This would seem to capture the dual nature of their discussion. But minimally we have no sense of the overlap of these current categories.

We appreciate your thoughts on this point. When drawing up our pre-registered analysis plan, we discussed in depth the benefits and drawbacks of modelling independent vs composite trajectories comprised of both life satisfaction and emotional difficulties. On balance, we feel independent trajectories yield more readily actionable results from a policy perspective. Moreover, given these are regularly cited as unique (albeit correlated) constructs for which a given predictor may influence one but not the other, we content independent models best capture sample heterogeneity and offer more nuanced findings.

That said, we recognise the large extent to which these models are currently siloed and that additional clarity surrounding cross-over between classes would offer better insight and inform the conclusions that can be drawn. As such, and in line with your suggestion, we now provide a cross-tabulation indicating the sample proportions for all possible combinations of class membership where participants’ are assigned to their most likely class (Page 12 / Table 3) and offer some discussion of cross-model proportions in the context of flourishing (Page 16-17).

  1. Clarity in the mediation model

Modeling mediation requires having a reasonable causal framework. While it is not fully clear what has gone into the deprivation score (see comment below) it still has to be argued that deprivation drives economy/work/employment in your model and that deprivation is not a result of economy/work/employment … there are other reverses or reciprocal specifications that could also be pointed out (e.g., housing/space/environment). This makes it difficult, in general, to understand the direction of the association from deprivation to neighborhood well-being measures and the decomposition of the direct and indirect effects of deprivation. The mediation concept is stretched here.  There may be more value (and consistent with their logic for the dual-state) by having just two latent constructs predicting MH classes. One being deprivation oriented (negative neighbourhood quality) and one being resilient oriented (positive neighbourhood quality).   This would also line up with more recent literature that argues that urban context not just be captured as disadvantaged but also by positive features (advantaged/resilient).

We agree that some components of neighbourhood wellbeing (housing affordability for example) also feature in the list of indicators used to derive IMD. For this reason, we are careful not to suggest a causal link between the two. Instead, we employ mediation as a framework within which we were able to tease out specifically, which components of neighbourhood deprivation may lead to better/worse mental health outcomes. All CWI items are continuous indicators of the socio-economic landscape of neighbourhoods hence, cannot be categorised as negative or positive. Rather, all are positively scored whereby higher scores reflect more optimal living conditions, lower scores reflect worse living conditions.

  1. The bias in the mediation model

In section 2.4.2. the authors explain they run the mediation model one intervening variable at a time.  They cite an issue of minimizing Type II error but each of these assessed “b” paths in their figure is highly biased by a set of left out variables. The magnitude of these effects will be highly related to the inter-correlations among the one variable of focus and the left out remaining variables. The c’ effect will likewise be biased and have differential size depending on deprivation’s differential relations to each of the well-being factors.  The only way this is not having an impact is if the correlations among the 9 factors of well-being are reasonably the same which is doubtful. But as it is their concern for Type II error is, in my opinion, rather trivial compared to the bias and how that bias may drive differences in the parameters of high interest to them. (I believe they would be better off just showing the result of their sensitivity analysis in detail).

Thank you for you insight here. We have now amended the manuscript given your concerns surrounding bias in the model. We now report and discuss results from models with all CWI items included simultaneously. Also please note, in accommodating your other suggestions regarding splitting life satisfaction and emotional difficulties into two tables and adding a cross-tab of cross-model class membership, the manuscript was inundated with Tables hence, to support readability, those denoting a and b-paths are now included as supplementary material. As these tables do not directly relate to our primary hypotheses, this adjustment does not affect the interpretability of our key findings.

  • Other meaningful covariates.

It is also difficult to assess some of the estimates without including some rather typical covariates in the models. Such things as gender(sex) or race/ethnicity are associate are rather basic to the assessment of MH. And such things as race/ethnicity might also be related to neighborhood location (a role of economic/social segregation).  If the individual characteristic is not controlled some of its effects/association may be captured by neighbourhood. Again, leading to potentially biased assessments of these key elements of interest.

We acknowledge the reviewer's concern regarding the omission of gender and ethnicity as covariates in our primary models. We continue to exclude these demographic covariates throughout enumeration of latent classes to align with best practice principles that ensure class formation is driven solely by the patterns of the indicator variables themselves. This approach mitigates the risk of covariates unduly influencing the classification solution, thereby preserving the interpretive clarity and theoretical integrity of the latent classes. However, in recognition of the relevance of these demographic factors, we have included results of adjusted models that include gender and ethnicity as individual-level covariates in the second stage of the analyses to provide insight into the extent to which these factors predict latent class membership as supplementary material.

Fixes/Needing Clarification:

  • The authors acknowledge a clustering by school but there seems little concern for geographical clustering or the hierarchical nature of the youth nested within the neighborhood.

Thank you for your suggestion here, we have now re-run analysis with models now incorporating a multi-level component that controls for neighbourhood clustering, in line with your suggestion. We appreciate your input here this ‘new and improved’ model is a substantial upgrade and better reflects the real-world structure of these variables. We describe the model building process in full on Page 9.

  • Table 3 and 4 could largely be removed and with the results of their choice of which class structure to believe in summarized and using Figure 3.

As requested, we have moved the tables containing model fit information from the main manuscript to Supplementary Material.

  • It is difficult to understand where the values for the neighborhood are coming from and their exact nature. This is especially the case for the IMD/deprivation index. It is not clear what goes into it.  Is it an aggregate measure derived from the youth or from some other source. If both the well-being and deprivation are summed qualities provided by the youth how accurate can their assessment of such things as employment or housing or equality be? Bottomline: it is very unclear where these neighborhood assessments are coming from… maybe the authors are assuming a greater familiarity with these ecological data than the typical reader.

We now provide further contextual information on the Community Wellbeing Index (CWI) and Index of Multiple Deprivation (IMD) on Pages 7-8. For reference, CWI were constructed externally by Co-Op and Geolytix – an external data analytics company and leverages a broad range of online centrally held data. IMD is the official measure of socio-economic deprivation in England, developed by the UK Government’s Office for National Statistics and linked to the #BeeWell dataset using participants’ home postcodes. IMD ranks correspond to Lower-Layer Super Output Areas (LSOA) which segment England in its entirety into 32,844 geographical units. For the current study, as in previous research, ranks 1 to 32,844 were converted to a percentage where higher scores represent areas of greater deprivation.

  • Table 7 is exceedingly dense. I would maybe remove the detailed variable label (table note them).  Or perhaps have one table for Satisfaction and one for Difficulties.

Variable labels have been abbreviated to simplify presentation with results for life satisfaction and emotional difficulties split into separate tables (Table 4 and 5). See Pages 14-15.

  • Also in table 7 … the “Chi-Square” [95% CI] is rather left unexplained. I’d prefer seeing the estimated coefficient of a*b … 

We have updated these statistics and labels and now provide the estimated coefficient of ab with explanatory footnotes.

  • I think it is better to label Phase I as identifying the classification model (see section 2.4.1.).

We have now amended ‘measurement model’ to ‘classification model’ throughout.

Citations…considerations.

  • Seems difficult to read a piece on neighborhood effects that doesn’t acknowledge the work out of Chicago’s neighborhood project by Robert Sampson, among others. Especially the different ways of measuring neighborhood factors and also approaches to testing.  An early review… “Assessing Neighborhood Effects: Social Processes and New Directions in Research” Annual Review of Sociology 2002(Vol 28) …Sampson, Morenoff, and Gannon-Rowley. But there are many others.

We now quote the work of Robert Sampson on The Chicago Project among others in an additional paragraph added to the introduction.

Reviewer 2 Report

Comments and Suggestions for Authors

Dear authors, it was a pleasure to read your article! Everything is very well presented, the logic is easy to follow, the methodology and results are thoroughly discussed, so I do not have any major comments. The only suggestion I have is to maybe provide a little more discussion on the finding that “neighborhood deprivation does not impact the likelihood of emotional difficulties”. 

Author Response

We thank the reviewer for their timely and insightful feedback. We believe the amends made according to each of your suggestions has vastly improved the quality of this piece of work. Please see our responses to each point below in red.

Dear authors, it was a pleasure to read your article! Everything is very well presented, the logic is easy to follow, the methodology and results are thoroughly discussed, so I do not have any major comments. The only suggestion I have is to maybe provide a little more discussion on the finding that “neighborhood deprivation does not impact the likelihood of emotional difficulties”. 

We have now expanded our discussion on this point on Page 17-18.

Reviewer 3 Report

Comments and Suggestions for Authors

The present research was conducted qualitatively, employing contemporary data collection methods.

The following observations may serve to improve the manuscript:

  1. The theoretical section requires further expansion through the incorporation of a new sub-section that explores the impact that one's place of residence has on mental health and psychological well-being. The utilization of a research model would be advantageous. The introduction is, as yet, extremely brief.
  2. The conclusions drawn in the section entitled “Conclusions” appear to lack the requisite rigor and substantiation. It is imperative that the authors provide a clear response to the objectives initially set out in the introductory section of the paper.

The article is perceived favorably on the whole.

Author Response

We thank the reviewer for their timely and insightful feedback. We believe the amends made according to each of your suggestions has vastly improved the quality of this piece of work. Please see our responses to each point below in red.

The present research was conducted qualitatively, employing contemporary data collection methods.

The following observations may serve to improve the manuscript:

1. The theoretical section requires further expansion through the incorporation of a new sub-section that explores the impact that one's place of residence has on mental health and psychological well-being. The utilization of a research model would be advantageous. The introduction is, as yet, extremely brief.

We have now included an additional subsection on neighbourhood effects in the introduction, citing work from the Chicago Project, among other sources.

2. The conclusions drawn in the section entitled “Conclusions” appear to lack the requisite rigor and substantiation. It is imperative that the authors provide a clear response to the objectives initially set out in the introductory section of the paper.

We have substantially amended the conclusions section to explicitly state how findings relate to our research questions while summarising key discussion points and policy recommendations (Page 20).

Round 2

Reviewer 1 Report

Comments and Suggestions for Authors

The authors have mostly satisfied my original discussion of the manuscript and have made substantial improvements.  My comments below are primarily clarifications and cleaning up of some small issues (unless I am misunderstanding something... by my misunderstandings should indicate some need to provide clarification).

Their discussion about flourishing is a nice add.

First, while the authors provide a cross-classification (Text Table 3) of the satisfaction and difficulties trajectories it doesn't fully address my  original comment to indicate the level of association between (among) the trajectories.  The distributions shown are nice but it still would be good to know how "correlated" the categories are.  Below I provide a crude example suggesting a fairly high level of association which has implications for the rationale of studying (or the ability to tease out) satisfaction as different from difficulties.  (Granted theoretically it makes sense to see them separately and the authors present that rationale just fine.... but if empirically they are highly tied together it makes their task difficult.) (See my work below the *********).

Second,  removing the tables to the supplementary materials is fine and works.  The one thing that would be useful to the reader is to point to the  "a" and  "b" results in supplementary table 3 and 4 when you begin to discuss  the a*b indirect effects (as you discuss Table 4 and 5 in the text) ... just to indicate many of the "a" coefficients are significant and to some extent warrant looking at the indirect effects (i.e. if none of these coefficients were related to the community variables than the notion of mediation would be pretty suspect---sort of a nod to the original Barron and Kenny view of mediation).  This just justifies why you focus on your indirect effects.

Finally...  review your supplementary table 3... there appears to be no table notes to explain the Life Sat and Emot Difficulties columns.  I'd also get rid of the "beta" CI since the Beta is really each alpha... confusing to mix the two.    And finally is it really the case these coefficients for alpha 1 through 9 almost exactly the same for Life Sat and for Emot Difficulties?  (I copied one row below).   This seems a bit odd and should be checked...)

Supplementary Table 3

a2

Equality

.268 * [.154, .382]

.268 * [.155, .382]

***************  crude look at the association between trajectories *************

Text Table 3 provides some help in understanding the joint distribution of the trajectories but part of the issue is the association among the trajectories.  This would help show that it is important to distinguish satisfaction (positive MH) from emotional problems (negative MH) (if the association is very high you would expect very similar predictors of both given the underlying model).  For example if one simply collapses consistently high and improving satisfaction as "good satisfaction" and collapse deteriorating and low into "bad satisfaction"... and likewise collapse low/subclinical as "OK emotion" and keep elevated as  "bad emotion" than you get a resulting 2 x 2 table (note I am unsure what to do with missing so I just ignore)

Good sat

Bad sat

OK emot

21111

416

Bad emot

 3470

743

Here if I know you have OK emotion the odds of having Good satisfaction are about  49 to 1... given OK emotion the probability of having good satisfaction is .98 (21111/21527) and the probability of bad satisfaction is .02  ---  odds of good satisfaction given OK emotion .98/.02 = 49... arguably a strong association.

Even if you throw sub-clinical into bad emotion you still get a pretty health OR of 6+.

Given some indication to the reader of the level of association or agreement among the trajectories would be helpful (i.e.  one could assume ordinal categories for each and calculate a non-parametric correlation between life satisfaction and emotional difficulties)... if really high it begs the question of treating them separately ... if moderate to low if supports the need to examine the outcomes separately with some expectation that the "independent community well being measures would likely (might) show different effects on satisfaction versus emotional difficulties.

Author Response

Comment 1: First, while the authors provide a cross-classification (Text Table 3) of the satisfaction and difficulties trajectories it doesn't fully address my  original comment to indicate the level of association between (among) the trajectories.  The distributions shown are nice but it still would be good to know how "correlated" the categories are.  Below I provide a crude example suggesting a fairly high level of association which has implications for the rationale of studying (or the ability to tease out) satisfaction as different from difficulties.  (Granted theoretically it makes sense to see them separately and the authors present that rationale just fine.... but if empirically they are highly tied together it makes their task difficult.) (See my work below the *********).

Response 1: We really appreciate your thoughts on this and worked example provided, this was really insightful. In response, as a compliment to information provided in Table 3 (n's per each combination of classes) we have added an additional table (Table 4) indicating the odds of following each pattern of trajectories and an explanatory paragraph explaining how these were calculated in line with your worked example. Indeed, there is cross-over between these constructs but some weak/moderate combinations and inconsistent dose-response effects (insofar as you can measure "dose-response" within this framework) supports the notion that they should be treated separately. Thanks again for this suggestion, we think this was a really helpful method of bringing findings together.

Comment 2: Second, removing the tables to the supplementary materials is fine and works.  The one thing that would be useful to the reader is to point to the  "a" and  "b" results in supplementary table 3 and 4 when you begin to discuss  the a*b indirect effects (as you discuss Table 4 and 5 in the text) ... just to indicate many of the "a" coefficients are significant and to some extent warrant looking at the indirect effects (i.e. if none of these coefficients were related to the community variables than the notion of mediation would be pretty suspect---sort of a nod to the original Barron and Kenny view of mediation).  This just justifies why you focus on your indirect effects.

Response 2: We now point the reader to the supplementary materials at the beginning of each section sub-titled "Indirect Effects Through Community Wellbeing" and note the significance of many a-pathways.

Comment 3: Finally...  review your supplementary table 3... there appears to be no table notes to explain the Life Sat and Emot Difficulties columns.  I'd also get rid of the "beta" CI since the Beta is really each alpha... confusing to mix the two.    And finally is it really the case these coefficients for alpha 1 through 9 almost exactly the same for Life Sat and for Emot Difficulties?  (I copied one row below). This seems a bit odd and should be checked...)

Response: We have added a table footnote highlighting, " * " reflects significant effects, updated the table headers to indicate life satisfaction and emotional difficulties columns provide results from the two separate models, and that coefficients are near-identical as the underlying samples providing data on the variables under investigation (neighbourhood deprivation and aspects of community wellbeing) were near-identical, save for those missing data on the main outcome for each. We have also removed "beta" CI to save confusion, in line with your suggestion.